# Photogrammetric reconstruction of 3D carpological collection in high resolution for plants authentication and species discovery

**Ho Lam Wang, Tin Hang Wong, Yiu Man Chan, Yat Sum Cheng, David Tai Wai Lau** *

Shiu-Ying Hu Herbarium, School of Life Sciences, The Chinese University of Hong Kong, Hong Kong, Hong Kong Special Administrative Region, The People's Republic of China

* lautaiwai@cuhk.edu.hk

**Data Availability Statement:** All relevant data are within the paper and its Supporting Information files.

## Abstract

This study provides an accurate and efficient method to reconstruct detailed and high-resolution digital 3D models of carpological materials by photogrammetric method, in which only about 100 to 150 images are required for each model reconstruction. The 3D models reflect the realistic morphology and genuine color of the carpological materials. The 3D models are scaled to represent the true size of the materials even as small as 3 mm in diameter. The interfaces are interactive, in which the 3D models can be rotated in 360° to observe the structures and be zoomed to inspect the macroscopic details. This new platform is beneficial for developing a virtual herbarium of carpological collection which is thus the most important to botanical authentication and education.

## Introduction

Mass digitization has become a trend in the past years to document natural history collections and herbarium archives which facilitated the knowledge transfer of biodiversity information [1–3]. 2D scanning is one of the major digitization methods in many herbaria and museums, in which specimens are captured as high-quality images [4–7]. However, 2D images are usually plain and fixed which cannot show a specimen from all angles of view. Also, some structures cannot be observed from the images due to overlapping [8].

Technological advancement has attracted much interest in the development of 3D digitization of natural history collections. The technique assists the transformation of natural history collection into 3D models which provides more information than those from the 2D counterpart. Laser scanning is one of the common methods to create 3D models of specimens with detailed external structures [9–11]. Computed tomography (CT) scanning is another method by using X-ray to penetrate specimens and document the internal structures in a pile of images, which can be stacked into 3D models [12, 13]. However, the genuine color of the specimens cannot be reflected on the 3D models reconstructed by laser or CT scanning, so this important phenotypic character is missing.

Photogrammetry has been applied in the 3D digitization of biological specimens by capturing images of the specimens from different angles. This method was firstly adopted on

**Funding:** The Native Plant Resources and Database in Hong Kong was established with the use of a donation from the Wu Jieh Yee Charitable Foundation. The virtual carpological herbarium of fruits and seeds described in this paper are a part of this database."

**Competing interests:** The authors have declared that no competing interests exist.

studying insect specimens [14–16], and later be extended to mammal and bird specimens [17–19]. The 3D models not only reflect the external structures of the specimens, but also reveal their color, which could aid for authentication and conservation of the animal species. Photogrammetry was further utilized to record live animals by multiple cameras to instantaneously capture images of the animals and then succeeded to the 3D model reconstruction [20]. Such approaches demonstrated how to digitize live species and natural history collections into 3D models by photogrammetry, which provide references for species documentation.

In the meantime, photogrammetry has been applied in various botanical studies. 3D models of live agricultural plants were reconstructed with photogrammetric methods, so as to estimate their growth rate and biomass, from which were the parameters were found to be highly accurate and precise [21–25]. Furthermore, photogrammetry also provided a low-cost and non-destructive alternative to document live plant records [26, 27]. Despite the wide application of digitization in animal specimens and agricultural plants, the photogrammetric reconstruction of plant specimens, especially their carpological parts, is still very rudimentary.

This research study outlined a tailor-made photogrammetric platform and a standardized methodology for digitizing carpological materials into 3D models. The settings are relatively stable and replicable. 3D models of 100 carpological materials had been reconstructed in this study. Results consolidated that this platform and method are feasible and replicable in mass digitization of carpological materials. The 3D models were also uploaded to a newly developed online database for researchers and the public. The limitations of this platform were also discussed with proposed solutions. In future, the new platform can be applied to create virtual carpological herbaria in different regions for the purposes of research and education.

## Materials and methods

### Trial usage of conventional 3D scanner

At first, an industrial 3D scanning platform consisted of a digital 3D scanner and a turntable was used to reconstruct 3D models of carpological materials. Although the reconstruction process was fully automated, the 3D models did not fulfill our research aims as it could not reflect the genuine color, resolution and structures of the carpological material. Hence, we began to build up our customized platform as below.

### Selection of digital camera body

In order to optimize the image quality and facilitate the image capturing process, several types of cameras were examined, including digital camera, digital single-lens reflex camera (DSLR) and mirrorless camera (Table 1). The digital camera Olympus TG-4 with a built-in microscope mode was the first one used in the experiment. Close-up images of insect specimens and small flowers were captured by using the TG-4, but the image quality could not be further analyzed because of their insufficient effective pixels (16 megapixels) and image resolution (4608 x 3456 pixels). Since the image resolution would significantly affect the quality of the 3D models, two

**Table 1. Specification of the camera bodies used in the experimental trials.**

|  | Olympus TG-4 | Canon EOS 80D | Canon EOS 5D SR | Canon EOS R5 |
|---|---|---|---|---|
| Camera type | Digital | DSLR | DSLR | Mirrorless |
| Effective pixels (megapixels) | 16 | 24.2 | 50.6 | 45 |
| Max. image resolution (pixels) | 4608 x 3456 | 6000 x 4000 | 8688 x 5792 | 8192 x 5464 |
| Touch screen | No | Yes | Yes | Yes |

DSLR cameras, namely Canon EOS 80D and EOS 5D SR were tried to achieve high image resolution up to 6K and 8K respectively. Both DSLR cameras were equipped with a touchable live-view panel, so as to minimize shutter-induced vibration. The reconstructed 3D models showed the macroscopic details of the carpological material. However, both DSLR cameras could not sharply focus on objects smaller than 5 mm in diameter in live-view shooting mode. The mirrorless camera, Canon EOS R5 with 8K-resolution was finally adopted because it performed better than the other two DSLR cameras, whereas it was found to provide high quality images of very small objects down to 3 mm under an efficient auto-focusing response.

## Selection of camera lenses

In the research of the magnification and image quality, three camera lenses were examined, including Canon MP-E 65mm f/2.8 1-5X Macro Photo, Canon EF 35mm f/2 IS USM and Canon EF 100mm f/2.8L Macro IS USM. The MP-E 65mm was expected to be better because of its 5 times magnification, however, the MP-E 65mm could not be operated by auto-focusing system, so its manual operation would largely increase the image capturing time. Hence, it was not practical and efficient for mass production of 3D models. The EF 35mm is a wide-angle lens with auto-focus mode, which could be used to capture images of larger carpological materials in good quality. It was noted that the EF 35mm has only a maximum magnification of 0.24 times which could not show the macroscopic details. The EF 100mm is a 1:1 macro lens with auto-focus mode that was found to be satisfactory for documenting both tiny and large carpological materials. Furthermore, the images under high magnification could still show very clear macroscopic structures of the materials. Finally, the EF 100mm macro lens was selected as the operational lens.

## Adjustment of light source

Lighting system of different configurations were conducted to ensure the 3D models reflecting the genuine color of the carpological materials with minimized image noise. Firstly, two lamps of no specification in light intensity and color temperature were placed over the carpological material. It was noted that the bottom part of the 3D model was darker than the upper part. Image noise was also relatively high because of ISO 6400 adoption under low light intensity of the LED lamps. Moreover, the 3D models could not show the genuine color because of unspecific color temperature. Hence, three light-plates of Phottix M200R RGB Light were used instead. The light intensity and the color temperature were adjusted to 100% and 5600K respectively to mimic the daylight spectrum. The light-plates were placed at both sides and the top, that facing directly towards the carpological material. The images captured and the 3D models reconstructed from this lighting settings could truly reflect the genuine color of the carpological material, and the image quality was very high because of minimized image noise.

## Software selection for photogrammetry and 3D model reconstruction

There are many photogrammetric software products available in the market. A number of trials were conducted to select a suitable one for researching our 3D reconstruction of carpological materials. The first completed 3D model in this study was a fruit of *Aleurites moluccana* (Linnaeus) Willdenow (PH005), reconstructed by using the AliceVision Meshroom (version 2021.1.0; https://alicevision.org/#meshroom). A total of 128 images were captured from four vertical angles towards the object (20˚, 0˚, -20˚ and -25˚) with a horizontal rotation interval of 11.25˚ until the turntable completely rotated for 360˚. 8K-resolution images were firstly used for reconstruction, but the 3D model was incomplete. It was likely because the size of the 8K-resolution images is too large that is out of the software analytical range. Hence, the images

were trimmed down to 4320 x 2880 pixels for matching the software capacity in order to reconstruct the 3D model. The whole experimental time was about three hours from image capturing to model reconstruction. The 3D model could show the shape and structure of the carpological material, but the model color was dimmer and the texture was blurred when comparing to the sample material. Additional trials of 3D model reconstruction were carried out by the AliceVision Meshroom, but the software processing time was too long to be favorable for mass production. On the other hand, image trimmed to lower resolution would lose some characteristic details that is not scientific for species authentication.

Afterwards, "3D Zephyr Free" (version 6.006; https://www.3dflow.net/3df-zephyr-free/), a trial version of the 3D Zephyr series was researched in the 3D model reconstruction, but it has a limitation of 50 images on each reconstruction analysis. Therefore, only 48 out of 128 images were selected, consisting three sets of 16 images from three vertical angles (20˚, 0˚ and -20˚) with horizontal rotation interval of 22.5˚. Unexpectedly, the reconstruction process finished within 30 minutes, and the quality of the 3D model was better than all previous models. They kept the genuine color and the texture, and showed the macroscopic details. Since the fruit of *A. moluccana* (PH005) had a relatively simple structure, further trials were conducted using a fruit of *Xanthium* sp. (PH003) with numerous of spines, to test the software capacity on analyzing more complex structures. 48 images of PH003 were captured using the same configuration, but the reconstruction was not successful. Possibly this was because PH003 was highly symmetric, and 48 images from three angles were not distinctive enough to be recognized by the software.

It was also noted that the number of images would directly determine the quality and the completeness of the 3D models, as well as the computing time of analysis. Importing more images to the software might provide more information for the reconstruction, which enhances the quality and the completeness of the 3D models, but the computing time would be much longer. For example, 3 days were required to analyze 700 images in some of our early trials. In contrast, importing less images to the software could reduce the computing time, but the 3D models might be incomplete or at low quality. Therefore, we have been experimenting on various image number in the new software to create the best quality of the 3D models with good efficiency.

The "3D Zephyr Lite" (version 6.006–6.010; https://www.3dflow.net/3df-zephyr-photogrammetry-software/)), an advanced version of the software with a maximum input of 500 images on each reconstruction was purchased in this regard. Basically, a total of 96 images of PH003 were captured from three vertical angles (20˚, 0˚ and -20˚) with a horizontal rotation interval of 11.25˚ were imported to the software. For more complicated materials, more images from wider vertical angles (e.g. 35˚ and -35˚) were required to produce a complete and high-quality 3D model. The procedures were repeatedly validated by applying in more than 50 carpological materials of different shapes and structures. The results further confirmed the usage of "3D Zephyr Lite" as software for photogrammetric reconstruction in our platform.

## Results

### New integration and establishment of standardized photogrammetric platform

After experimentation and optimization of the equipment settings and their usage, an image capturing platform and a standardized workflow for 3D model reconstruction were established, which enable an efficient 3D digitization of carpological materials. This platform has successfully reconstructed 3D models of 100 carpological materials. The details of the platform and workflow are described as follows.

## Apparatus and configuration

The settings of the image capturing station are shown in Fig 1. A mirrorless camera Canon EOS R5 with a macro lens Canon EF100mm f/2.8L Macro IS USM using a mount adapter EF-EOS R was used to capture images of carpological materials. The camera was fixed on a tripod, consisting of a Manfrotto 190D stand, a Manfrotto XPRO geared three-way pan/tilt tripod head, and a Fotomate LP-01 macro-turning long-type rail slider. A Phottix tabletop portable photo studio was used to provide a matte background of either white or black to distinguish the carpological materials of different colors.

The carpological material was attached firmly on a pin with diameter corresponding to the material size. The pin was fixed in vertical orientation on a plastic container. A tailor-made triangular marker was affixed through the pin as a size reference. The lateral faces of the marker had scale rulers and specific herbarium logo which aid for accurate image recognition by the software. The carpological material with the marker were then placed at the middle of a Com-Xim electric turntable MT200RL20, which enabled image capturing from 360˚ view. Three

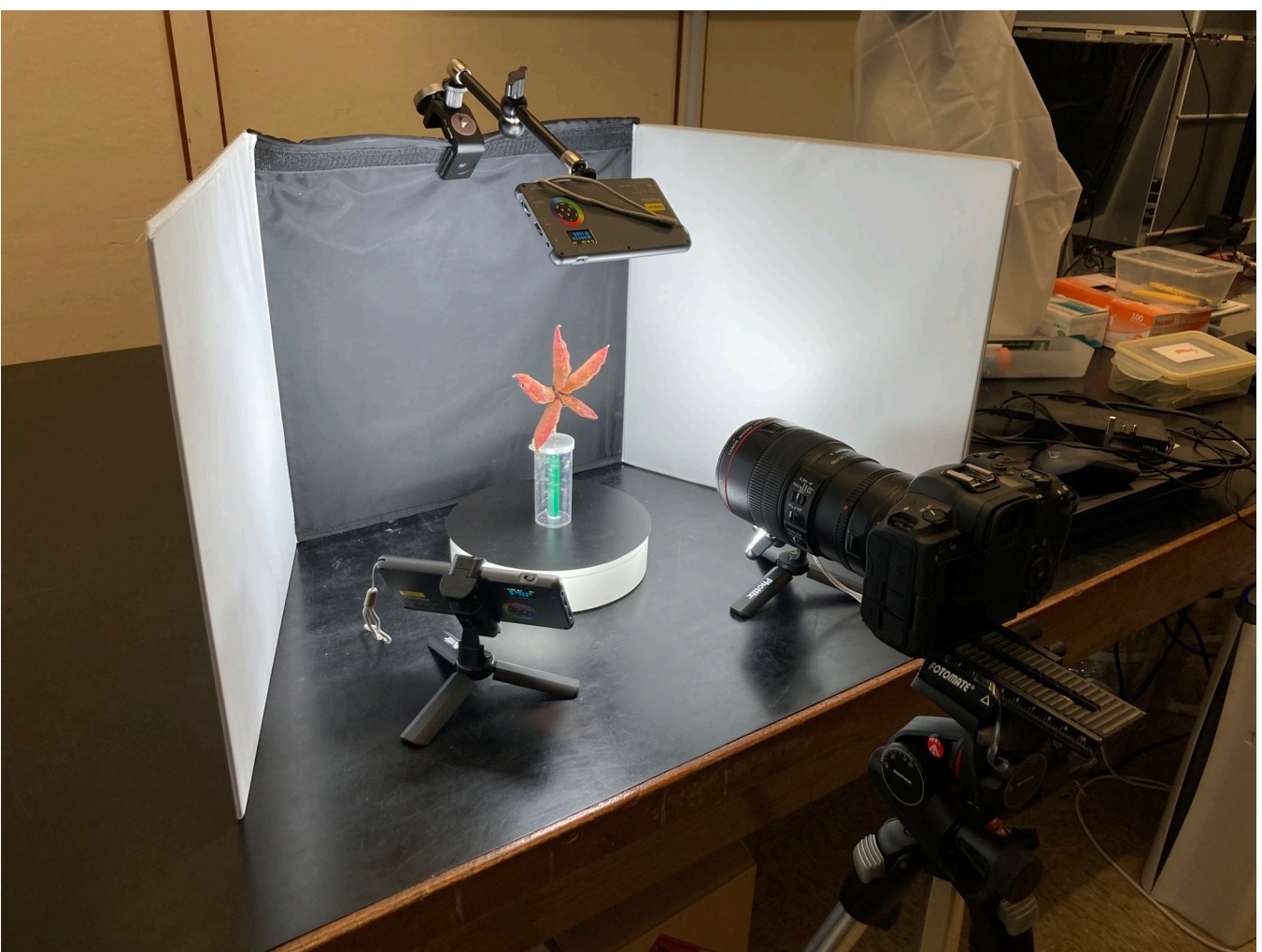

**Fig 1. Set up of the photogrammetric station.** A mirrorless camera Canon EOS R5 with a macro lens (Canon EF 100mm f/2.8L Macro IS USM) is mounted on a tripod. The carpological material is affixed on a pin which is placed in the middle of a turntable. Three light-plates are placed on the top, left and right side of the carpological material. A portable photo studio is used as background.

**Table 2. Configuration of the camera Canon EOS R5 for image capturing.**

| Camera mode | Manual |
| --- | --- |
| Shutter speed (s) | 1/100 |
| Aperture | f13 |
| Light sensitivity | ISO 800–2000 |
| Color temperature (K) | 5600 |
| Image resolution (pixels) | 8192 x 5464 |

Phottix M200R RGB Light were installed on top, left and right sides of the carpological material from a distance of 15 to 20 cm, so as to provide stable and well-dispersed light source.

## Image capturing

The settings of the camera were kept consistent throughout the image capturing procedures. The adopted configuration was shown in Table 2. The aperture was fixed at f/13. The light sensitivity (ISO) of the camera was adjusted according to the morphological structures and the color of the carpological material. The settings of the light-plates were also remained consistent (CCT mode; color temperature 5600 K). The camera and the macro lens were adjusted to auto-focus mode and the images were captured in live-view shooting mode.

The images were captured basically from three vertical angles to the carpological materials (any angle between 20˚ to 25˚, 0˚ and any angle between -20˚ to -25˚), with a horizontal rotation of 11.25˚ until a cycle was completed. Therefore, a total of 96 images were captured for each reconstruction (Fig 2). For carpological materials of very complicated structures, additional images from vertical angles 35˚ and -35˚ were captured to acquire more information of the carpological material. At each angle, the camera was adjusted to the minimum focusing distance from the carpological material, or until the carpological material and the marker appeared in full size on the camera monitor by moving the rail slider. It is important to capture the carpological material together with the marker in order to provide a real-time reference of the material size.

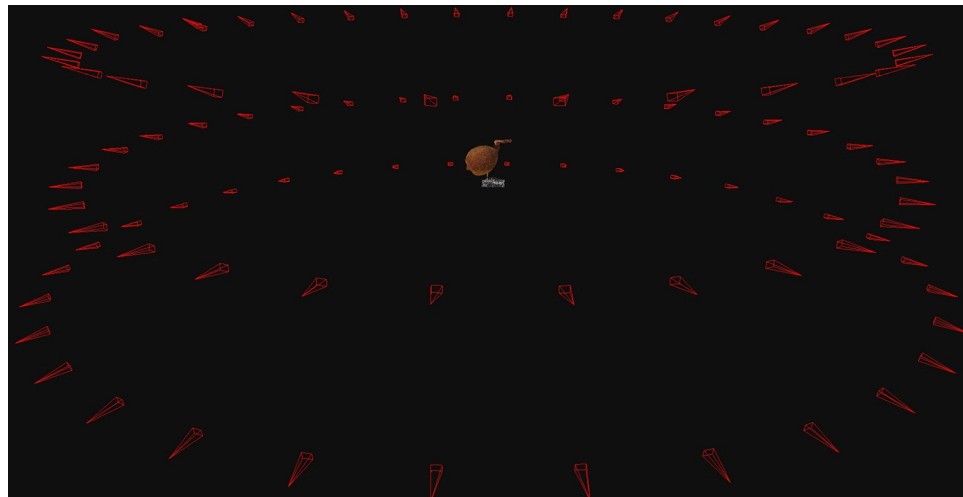

**Fig 2. Camera position estimated by 3D Zephyr Lite.** 97 images of the fruit of *Ficus pumila* Linnaeus (PH013) were captured from 0˚, 20˚, and -20˚ vertically towards the carpological material, with at a horizontal rotation interval of 11.25˚. The red triangles represent the camera positions estimated by 3D Zephyr Lite.

## Image analysis and 3D model reconstruction

The images were analyzed by a software 3DF Zephyr Lite under a high-ranked computing system (CPU: Alienware Aurora R11 with an 11<sup>th</sup> Gen Intel® Core ™ i7 11700KF; GPU: NVIDIA GeForce RTX 3090). For the software settings, the category and presets for camera orientation were "General" and "Deep" respectively. "General" and "High details" were selected for dense point cloud creation, surface reconstruction, and texturing. The software would show the number of selected images for the model reconstruction after internal calculation, and the data point cloud of the 3D model and the positions of image capturing would be previewed.

The reconstructed 3D model consisted of three major components, namely "Mesh" and "Texture" (Fig 3), and a MTL file. Mesh referred to a monochrome solid model composed of many polygons, while texture refers to the surface color of the model. The texture was exported as a separated JPG/PNG file after the reconstruction. A MTL file would also be created to command the application of texture to the mesh automatically. Hence, the 3D model was composed of a mesh covered by a layer of texture.

## 3-D model editing

During model reconstruction, some unwanted meshes such as the background and the pin would also be reconstructed. Therefore, the 3D model was exported as textured mesh and imported to another open software Blender (version 2.93; https://www.blender.org/) for manual editing. The unwanted structures could be deleted while the specimen with the scale bar were kept in the 3D model. The edited 3D model was saved as a new file of the final product.

## Establishment of "Virtual Carpological Herbarium of Fruits and Seeds" in the Shiu-Ying Hu Herbarium

In this study, 3D models of 100 carpological materials were reconstructed (S1 Table). Each 3D model was assigned with a code (e.g. PH001) for archival record in herbarium. The 3D models PH001 to PH055 were derived from carpological materials during the method development stage, while 3D models from PH056 onwards were fresh carpological parts of newly collected and authenticated specimens. The successful reconstruction of these 100 models had further consolidated a complete workflow of the photogrammetric platform (Fig 4). All the 3D models were uploaded to an open online database (https://syhuherbarium.sls.cuhk.edu.hk/collections/3d-specimen/; Username: syhuherbarium; Password: @CUHK). Currently, over 250 3D models were uploaded to the online database and more will be released in the future.

The size of the carpological materials varied greatly, ranging from about 3 mm to 12 cm in length. They were diverse in fruit types, including nuts, aggregated follicles, schizocarps, drupes, capsules, legumes, berries etc. The well-established standard operation procedures and the 3D models significantly enabled the establishment of a virtual carpological collection. With a user-friendly model viewer, a public database namely "Virtual Carpological Herbarium of Fruits and Seeds" (https://syhuherbarium.sls.cuhk.edu.hk/collections/3d-specimen/) was established for archiving virtual carpological collection. Since the database is a responsive webpage, the interfaces can function properly in various kinds of mobile devices or computers. The 3D models can be viewed with a HTML5 browser without any software installation in the user devices. All 3D models in the database can be freely browsed in 360˚ by simple mouse gestures on desktop computers or by fingers on mobile devices. In order to browse the virtual specimens smoothly, some hardware and software configurations are highly recommended:

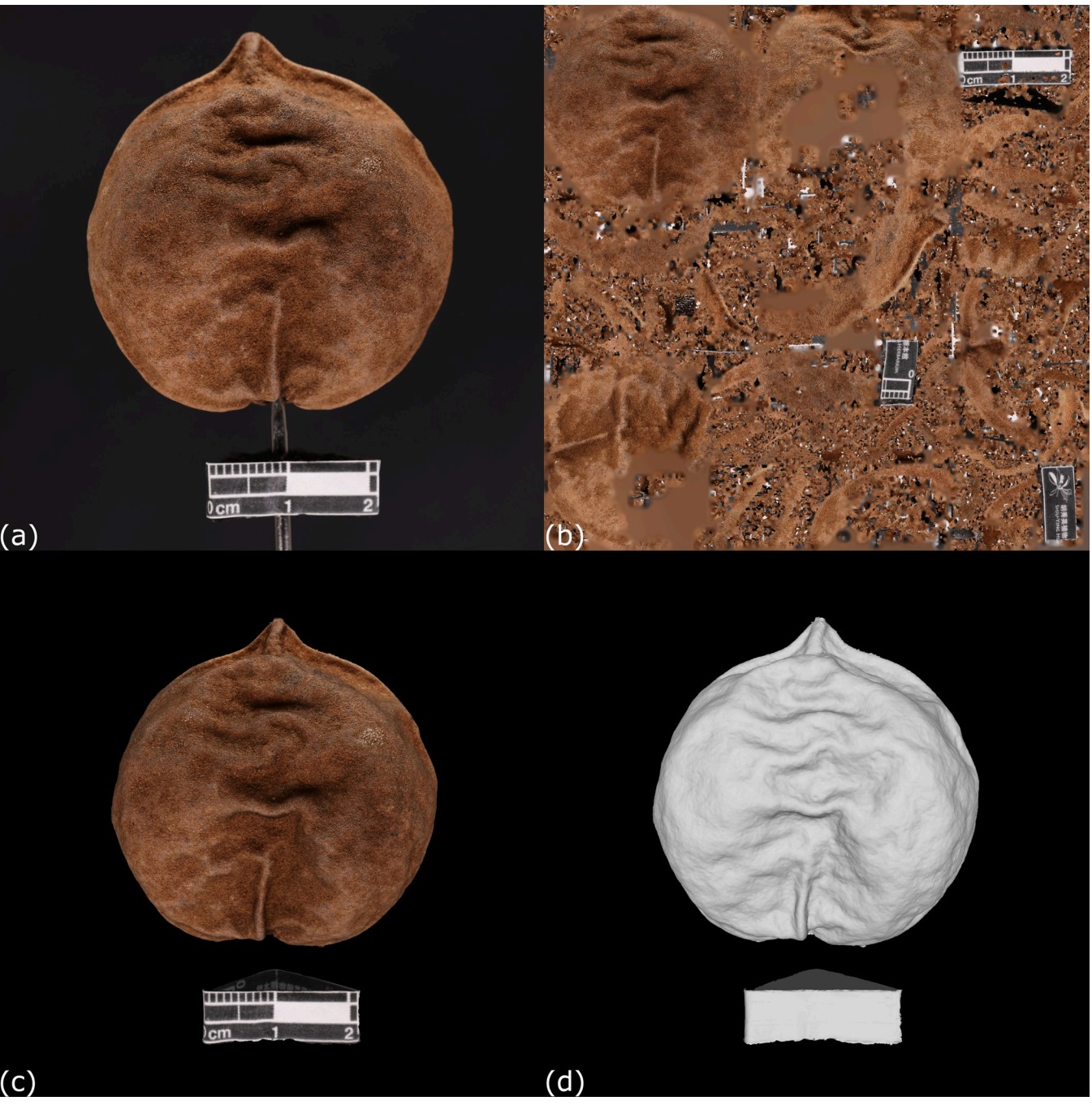

**Fig 3. Composition of a 3D model reconstructed by photogrammetric platform.** (a) A carpological material of the fruit of *Aleurites moluccana* (Linnaeus) Willdenow (PH005). (b) Texture of the 3D model extracted by 3DF Zephyr Lite. (c) A 3D model with texture to show the genuine color of PH005. (d) A 3D model without texture to show the plain mesh of PH005.

For desktop or laptop computers, CPU: Intel i5 2.1G or above; RAM: 6G or above; 3D GPU: CUDA enabled graphics card (e.g. Nvidia GeForce 1650 or above); Web browsers: Chrome 93.0.4577.82 or above / Firefox v92.0 or above / Edge v93.0.961.47 or above / Safari v5.1.7 or above were suggested. For mobile or tablet, OS: Android 9 or above/ iOS 14.8 or above were suggested.

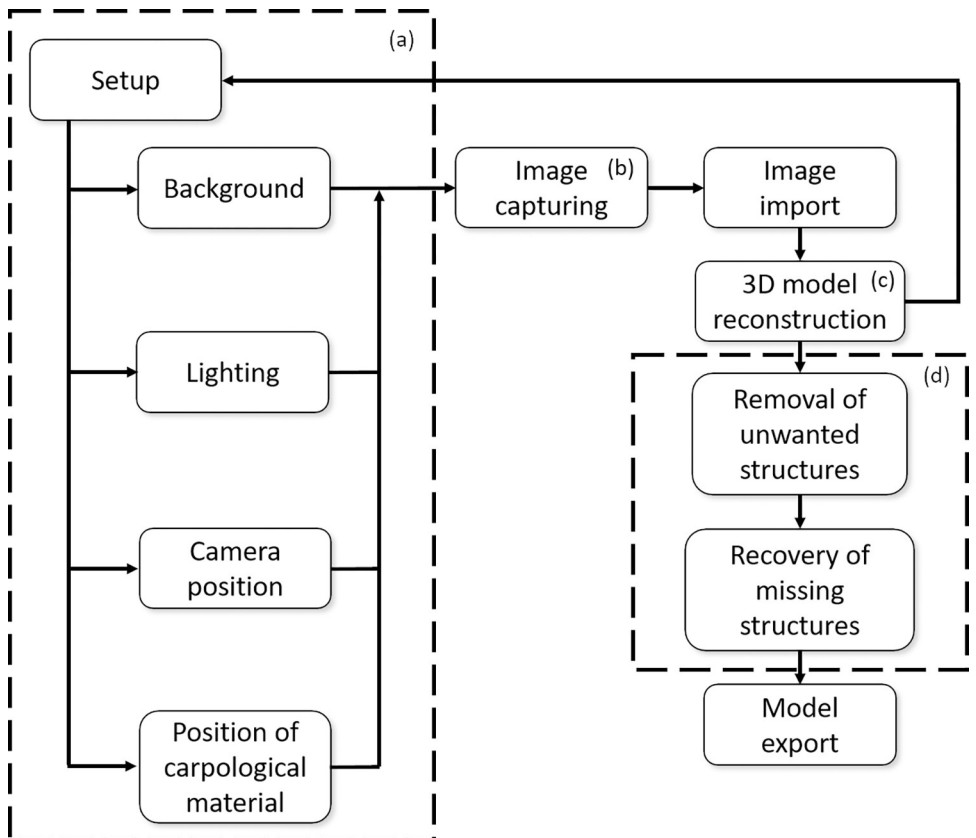

**Fig 4. A workflow for 3D digitization of carpological material.** (a) The set-up of the image capturing system includes background color, lighting, camera position and the position of the carpological material. The adjustment of these parameters depends on the morphology of the carpological material. (b) Image capturing process of each material usually lasts for 5 to 10 minutes. (c) The images are imported to the software for 3D model reconstruction. The set-up should be adjusted if the reconstruction was failed. (d) Post-reconstruction editing would be carried out if the model is successfully reconstructed.

## Discussion

### Advantages of the photogrammetric platform and the derived 3D models in this study

All the 3D models reconstructed by the photogrammetric platform were diverse in fruit types. Each has its unique morphological structure which can be observed clearly from all angles, such as the prickles on the fruit of *Xanthium* sp. (PH003) and the fruit of *Castanopsis lamontii* Hance (PH019 & PH045), the persistent involucre on the fruit of *Passiflora foetida* Linnaeus (PH034), and the grooves on the fruit of *Alpinia* sp. (PH050). These results demonstrated that the complex structures of carpological materials could be reconstructed by this photogrammetric platform. Most of these are important phenotypic traits for species identification and classification. Therefore, these 3D models could be used as important references in science and education.

All the 3D models have been uploaded to an online database, not only limited to herbarium staffs but also open to co-workers and the general public. Users can freely inspect the 3D models of the carpological material by simply dragging on the interface. Since a macro lens was used to capture the images, most of the macroscopic details could be clearly shown even when the 3D models were magnified up to 10 times. The structural fine details in 360° view would

enable the users to inspect the 3D model that looks almost identical to the real carpological materials.

Instead of adding a scale bar during the post-reconstruction stage, a customized scale bar was attached right below the carpological material during the image capturing stage as a reference of their realistic size, so all the 3D models include a carpological material and a scale bar which enabled an accurate size determination. Moreover, the photogrammetric platform is able to reconstruct carpological materials of wide range of size such as the seed of *Embelia* sp. (PH021) in 3 mm and the fruit of *Samanea saman* (Jacquin) Merrill J. Wash. (PH044) in 12 cm.

The color temperature of the light-plates was adjusted to 5600K which is identical to sunlight and the settings of the image capturing procedures were kept consistent. Hence, the final 3D models derived from the software reconstruction would not have color deviation, and truly reflected the genuine color of the specimens. Therefore, the 3D models of high color accuracy would be beneficial for species authentication.

Conventionally, it might be time-consuming and costly when scientists need to visit a distant herbarium for specimen inspection. Specimen loan could be helpful, but probably increase the risk of specimen damage. Currently, most virtual herbaria have partially resolved the problems by providing 2D images of the specimens. However, the new photogrammetric platform in this study could revolutionize a new documentation of the carpological parts such as fruits and seeds which are crucial in authentication. Hence, the 3D model database could facilitate the usage of herbarium specimens across different regions and generations.

This platform has provided an efficient documentation of massive carpological collections. Only about 100 to 150 images were required to reconstruct a high-resolution 3D model. The file size of each 3D model reconstructed in this study (including texture PNG, mesh OBJ and MTL file) was between 22.6 MB to 225.6 MB depending on the size and the complexity of the carpological materials. The size of the PNG file and the OBJ file was between 13.7 MB to 135.6 MB and between 8.9 MB to 31.4 MB respectively. The size of the MTL file was only 1 KB for all 3D models as it was just simple coding. During the experiment, the time required for image capturing was about 10 to 15 minutes for each carpological material, whereas the time for model reconstruction and editing required approximately an hour to complete. The long-term aim of this photogrammetric platform is to document the carpological parts of over 2,000 native plant species in Hong Kong within a 5-year period. The cost of all equipment was about USD$6,000 in total and the price of the perpetual license of the software was about USD$160. Including the labor cost for manual operation, the estimated cost of each 3D model in this research study is about USD$25. As this method is relatively low cost and easy to operate, it is practical for a typical herbarium to customize a photogrammetric platform for 3D documentation of carpological materials.

## Resolved problems in this research

Light reflection from the material surface was the major problem during image capturing. The white spots reflected from light source could impair the image recognition by the software, and resulted in reconstruction failure. Nonetheless, this problem could be fixed by adding various kinds of diffusers such as white paper or soft box to diffuse the light towards the carpological materials (Fig 5).

On the other hand, the specimen orientation could significantly affect the result, and sometimes lead to reconstruction failure, especially in the long carpological material. The fruit of *Samanea saman* (Jacquin) Merrill J. Wash. (PH044) was used to illustrate the situation (Fig 6). Two sets of images were captured in horizontal and vertical orientation respectively. Results

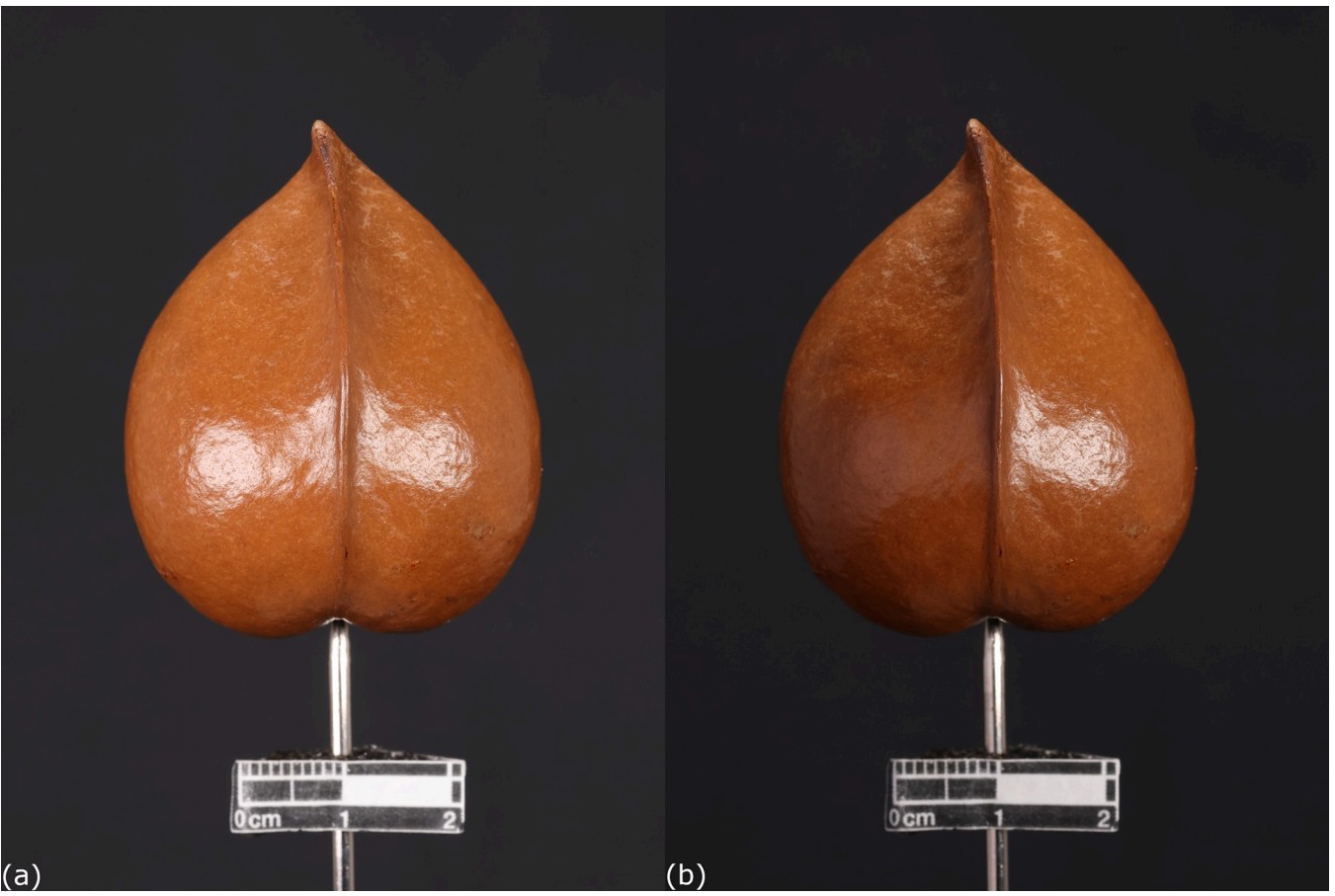

**Fig 5. Light reflection on shiny surface.** (a) Two light plates were placed on the left and right sides of the fruit of *Heritiera littoralis* Aiton (PH018) respectively. The light reflection created multiple white spots on the surface which resulted in failure of image recognition by the software. (b) A paper was added between PH018 and the left light plate, which diffused most of the direct light towards it. The light reflection on the left side was significantly reduced.

showed that only one side of PH044 was reconstructed when it was placed horizontally. The image depth of field could be a reason, because it was not feasible to get clear images from both ends of a long carpological material in horizontal position. Furthermore, the software might not be able to distinguish both ends of PH044 due to blurred image. A 3D model of long carpological material could be successfully reconstructed when it was placed in vertical orientation, as the images are clearly enough to be recognized by the software (Fig 6).

## Problems required further investigation

Although the results demonstrated that photogrammetry can be used to reconstruct 3D models of carpological materials, certain structures could not be reconstructed yet. Firstly, it was observed that the indumentum such as hairy or scaly coverings of fruits or seeds could not be reconstructed precisely. These structures can still be observed on the 3D models when the texture is applied. However, once the texture is turned off, these structures become less recognizable on the pure mesh (Fig 7). One possible reason was that the software misinterpreted the indumentum as flatten patterns on the surface instead of their tiny outgrowth structures, which merged them together as a single surface. As a result, the indumentum could not be reconstructed by the platform precisely. Similar problems were found in reconstructing 3D

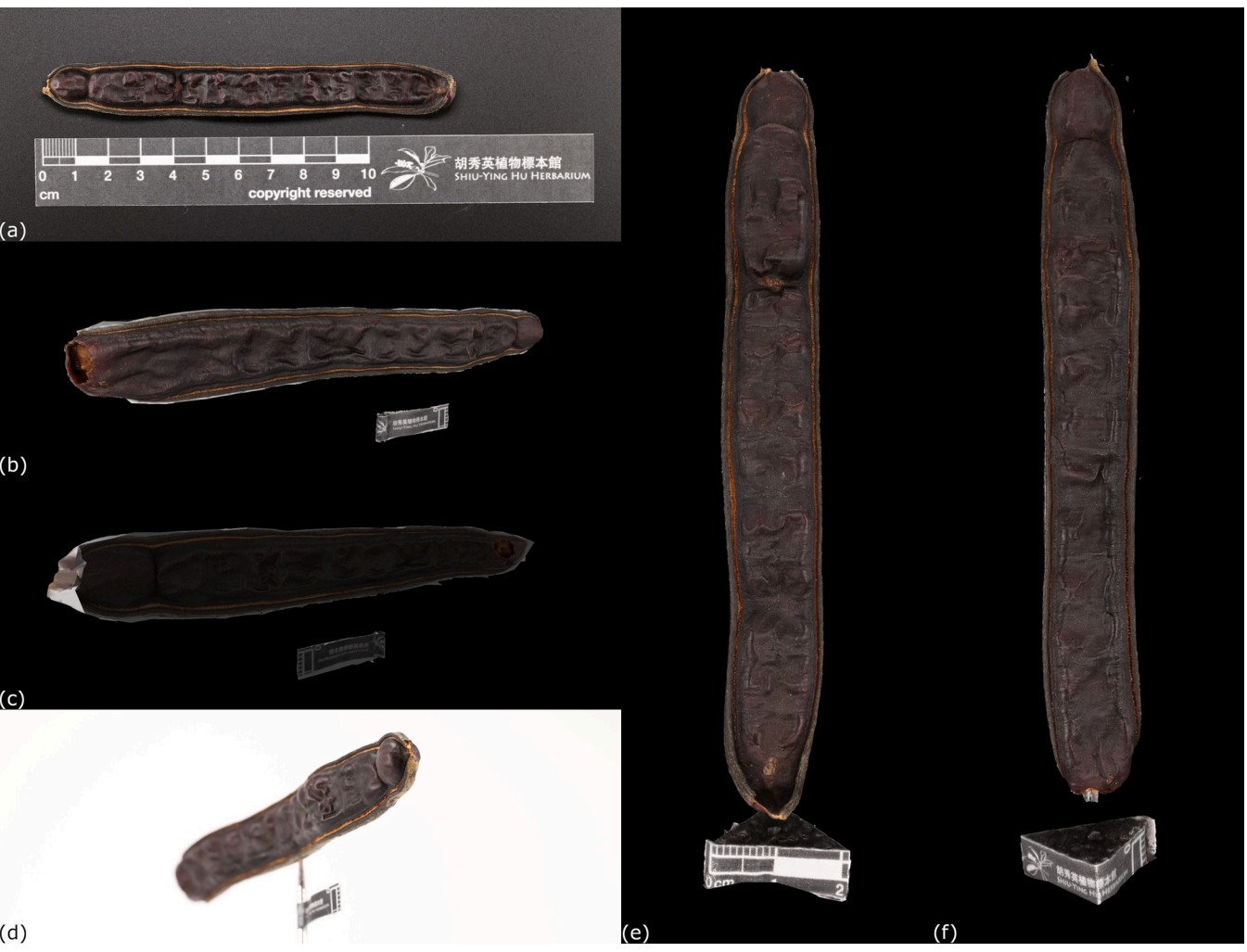

**Fig 6. Comparison of 3D models of long dimension carpological material reconstructed by different positioning.** (a) A carpological material of the fruit of *Samanea saman* (Jacquin) Merrill J. Wash (PH044). (b)&(c) The 3D model when PH044 was placed horizontally. The structure on the front side was reconstructed but the back side was not, as shown by the regions with darker color. (d) Decrease in sharpness of the images of PH044 was found due to greater depth of field from two ends, which reduced the coherence between images so the software could not recognize its structure for further reconstruction. (e)&(f) The 3D model when PH044 was placed vertically. The structure was reconstructed completely on both sides.

models of birds or insect specimens by photogrammetry, in which the feathers or hairs were lost, or merged with the surface structures [16, 19].

Secondly, long filament-like, very thin, and light structures could not be reconstructed by this platform. Multiple trials were conducted to reconstruct a 3D model of carpological material with filament-like structures (Fig 8). The results showed that the software was not able to recognize the position of the camera which led to unsuccessful reconstruction. We believed that the small changes in physical positions of the coma due to rotation or air ventilation would result in discontinuity of the image recognition. Hence, the software treated the images as from different objects but not from the same one. Therefore, it cannot combine all of them for 3D model reconstruction.

Thirdly, semi-transparent structures could not be shown on the 3D models in this study. Instead, that parts would be stained with some background color or become a solid color, mainly because the software could not distinguish whether the parts are transparent or not, and simply adopted the background color that passing through the tissue (Fig 9). As a result, the

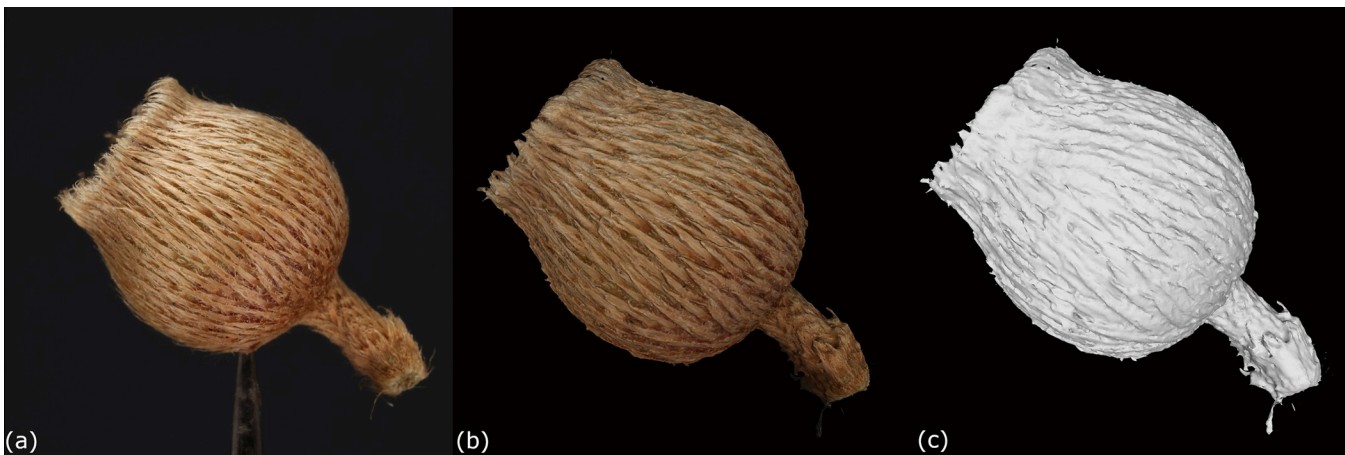

**Fig 7. A comparison of the indumentum of 3D models and the carpological materials.** (a) Side view of a carpological material of *Melastoma malabathricum* Linnaeus. (b) Side view of the 3D model PH193. (c) Side view of the mesh with texture turned off.

reconstructed 3D model could not demonstrate the genuine texture and color of the material in different degree of transparency. This problem also occurred in 3D models of insect specimens which the color of the abdomen appeared on the wings due to transparency [14].

**Fig 8. Reconstruction trials of carpological materials with filament-like and thin structures.** (a) A Seed of *Gymnema sylvestre* (Retzius) Schultes with coma attached. (b)to(f) Results of the reconstruction trials of *G. sylvestre* showing the positions of the camera estimated by 3D Zephyr and the 3D models.

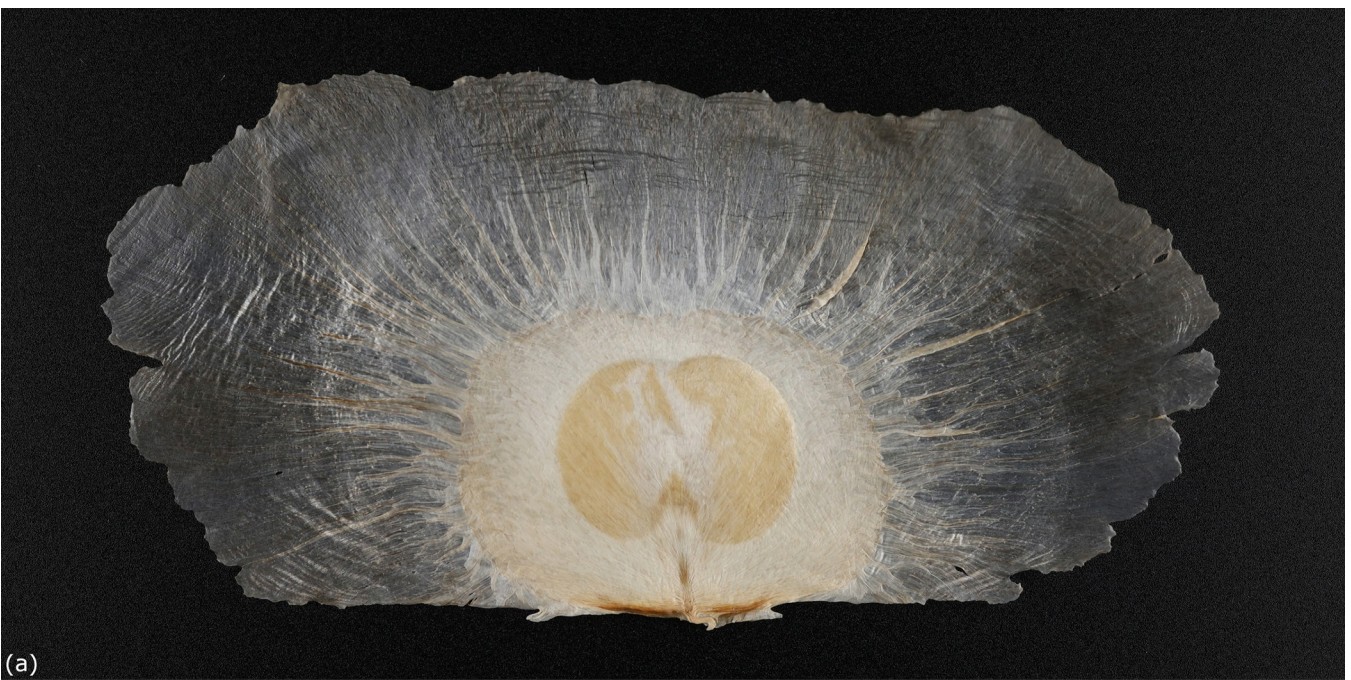

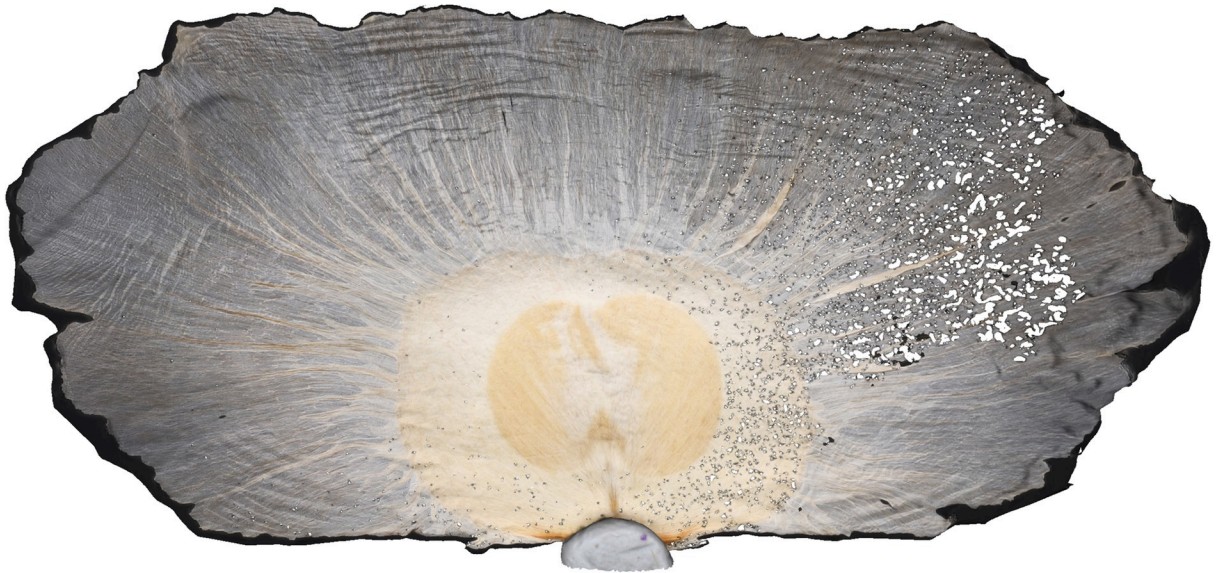

**Fig 9. Reconstruction trials of carpological materials with transparency.** (a) A seed of *Oroxylum indicum* (Linnaeus) Bentham ex Kurz. The background color passed through the transparent parts of the seed. (b) A 3D model of the same seed reconstructed with a black background. The transparent parts were stained with background color.

## Comparison of the new platform to other competing techniques

**2D specimen images.** Both 2D scanning and the photogrammetric platform in this study document the specimens in digital images which would not exhibit physical or biological degradation as happened in the real specimens. However, images captured by 2D scanning only show a single side of the specimens which might not show some important characteristics.

Multiple 2D images taken from different angles were required to deduce the real morphology of the specimens. Even if so, the observation was limited to only a few angles, which could be confusing when the specimens are complex in structure. The photogrammetric platform in this study aims at documenting the reproductive structures of a plant, mainly the carpological parts, in 360˚ for authentication. The carpological materials can be observed at any angles as the 3D model can be rotated freely. Hence, the 3D models could provide additional information than those from the 2D images. Last but not least, only a single 3D model file is required instead of multiple photos, hence it greatly facilitates the efficiency of storage and management.

**3D scanners.** Most of the 3D scanners available on the market are bundled with default cameras, scanning platform and editing software, from which the equipment settings are fixed and not replaceable. They are designed for scanning objects within certain size range, usually ideal for objects ranging from 3 cm to 25 cm. Therefore, the compatibility and sustainability of these 3D scanners are limited. In contrast, the software and hardware in this photogrammetric platform are independent components and replaceable, so it allows further upgrade of any components. On the other hand, the settings adopted in this study are able to reconstruct carpological materials from 3 mm to 12 cm in diameter, which have included a wide size range of fruits and seeds. In addition, this platform does not require calibration before image acquisition, but most of the 3D scanners do. Therefore, the photogrammetric platform and workflow in this study are more applicable and user-friendly.

**CT scanning.** CT scanning could reconstruct the specimen internal structures with numerous images, which can be stacked into 3D models, but they do not have colored surface and cannot reflect true phenotypic characteristics. Ijiri *et al.* [28] demonstrated a combination of CT scanning and image-based approach to create 3D models of insects and flowers with both internal structures and colored surface textures. However, the methodology required multiple steps and professional operations. In this study, the 3D models derived from the platform could also reflect the genuine color and phenotypic structures of the carpological materials, and the entire procedures are practically easier.

**Other photogrammetric methods.** Currently, a few 3D models of plant carpological parts by photogrammetric reconstruction were found on a public platform named Sketchfab (https://sketchfab.com/disc3d/models), but massive database of digitized plant carpological materials was not yet found. On the other hand, more studies have been conducted in reconstructing 3D models of agricultural species to measure their size and growth rate. Therefore, the photogrammetric platform of this study was carpological materials specific, and it also provided 3D models with detailed morphology. Moreover, it was demonstrated to be user-friendly to reconstruct high quality 3D models from about 100 to 150 images.

**Photometric stereo.** Photometric stereo is a technique that capture successive images at the same viewing direction while varying the light direction towards the object, hence to determine the surface orientation [29]. It has been applied in the reconstruction of 3D models in different studies [30–34]. This method requires less images to create surfaces with more detailed topography than photogrammetry, but there are several constraints when it comes to the 3D reconstruction of carpological materials.

Firstly, photometric stereo involves complicated mathematical calculations. Users need to input assumptions, algorithm and formula before 3D reconstruction which requires proficient knowledge in mathematics and engineering. It could be difficult for a botanist to operate such calculations and may not be practical in a typical herbarium.

Secondly, photometric stereo requires the conversion of colored images into gray-scale before reconstruction, hence the resulted 3D surface is also gray-scale. Color photometric stereo is a modified approach by illuminating lights of different colors towards the objects, yet the

resulted 3D surface still cannot reflect the genuine color of the objects [34, 35]. Therefore, the genuine color of the carpological materials will be lost which is not suitable for species authentication and identification.

Thirdly, we observed that the objects examined in photometric stereo studies mainly consist of a single substance which means they have the same reflectance throughout. However, carpological materials usually include multiple parts such as fruit, calyx and pedicel, which they all have different texture and different reflectance. Since there is no studies on utilizing photometric stereo approach to reconstruct 3D models of carpological materials or any plant parts, the uncertainty of photometric stereo still exists. Meanwhile, improper assumptions could also lead to global deformation and shape inaccuracy of the 3D models [36, 37].

In contrast, the photogrammetric method in this study is relatively simple which requires less proficient knowledge in mathematics and engineering. The 3D reconstruction requires single-step operation by the software even the carpological materials consist of multiple texture. The reconstructed 3D models reflect the genuine color of the carpological materials which is beneficial for species authentication and identification. Hence, this photogrammetric method is more practical to a typical herbarium.

**3D sculpting.** Compared to other mesh generation approaches of manual manipulation including sculpting and polygon drawing, the platform of this study serves the purposes of scientific documentation. Manual manipulation methods require sophisticated techniques in 3D modelling to create a lifelike model, but it does not sufficiently reflect the genuine features of the specimens. The artificial additions of characters would require detailed verification by professionals along the production. Hence, large-scale production of 3D models by sculpting techniques would be time-consuming and costly.

## Application of 3D models in the improvement of plant knowledge

Every specimen is a paramount scientific evidence for species identification and discovery, but damage and deterioration of the specimens could impair their authentication value, especially in type specimens. Specimens of over 100 years would probably have deteriorated morphology. Moreover, deformation and shrinking of some specimen parts are inevitable during the preparation procedures, and the specimens color also change over time. Hence, the specimen users could not differentiate which specimen characters are genuine or fake due to structure degradation. In contrast, 3D models reconstructed in this study could overcome these shortcomings and preserve the original form of the carpological materials in a digital version, which can be demonstrated by the following examples.

A local plant (*Sterculia lanceolata* Cavanilles) with follicle-type fruits was selected to explain this situation. Its specimen of over 50 years (Shiu Ying Hu 5569) and a 3D model of its fresh fruit (PH053) were compared (Fig 10). The general description of living *S. lanceolata* fruits is red to bright scarlet in color, long ovate or ellipsoid, densely pubescent, base attenuate and apex beaked [38, 39]. However, some of these characteristics were lost in the old specimen (Shiu Ying Hu 5569). The color of the specimen has already faded out. The fruit surface has a harder texture and numerous wrinkles. The changes of these features would give inaccurate evidences in authentication. Alternatively, a 3D model of a freshly matured fruit of *S. lanceolata* (PH053) was reconstructed shortly after collection. The morphology of the fruit was well recorded in the 3D model that matched with the major characteristics including color and follicle shape as mentioned in Floras. Therefore, a good documentation of these 3D models in a database could be essential to botanists.

Sometimes, the descriptions of plant carpological parts maybe ambiguous, too brief, or even absent in Floras, which can hardly reflect the morphology of these reproductive features.

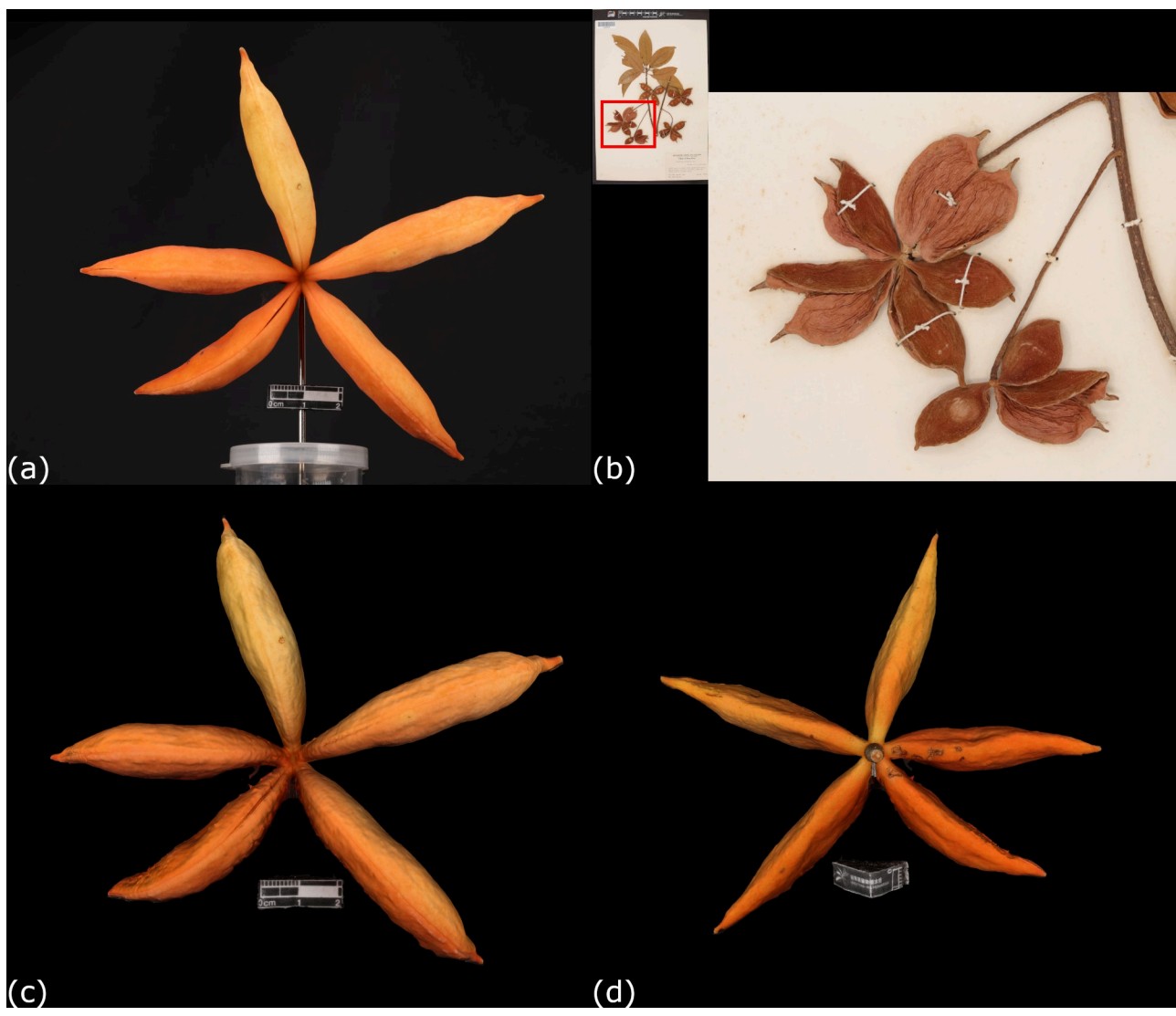

**Fig 10. Comparison between 2D image and 3D models of fruits of *Sterculia lanceolata* Cavanilles.** (a) A carpological material of the fruit of *S. lanceolata* (PH053). (b) 2D image of a specimen of *S. lanceolata* (Shiu Ying Hu 5569). (c) Front side of the 3D model of PH053. (d) Back side of the 3D model of PH053.

However, with the aids of 3D models, the entire morphology of the carpological parts can be visualized for better understanding. Four 3D models (PH223, PH224, PH267 & PH268) of the fruit and seed of two local plants *Daphniphyllum oldhamii* (Hemsley) K. Rosenthal and *Daphniphyllum calycinum* Bentham were reconstructed in fresh after collection. In Flora of Hong Kong, the carpological parts of *D. oldhamii* is described as "Drupes ellipsoid or obovoid, 8 x 6 mm, base without persistent calyx lobes", while *D. calycinum* is described as "Drupes ovoid, ca. 1 cm, pruinose, papillose, base with persistent calyx lobes" [40, 41]. The major difference of their fruits, which is the absence or presence of persistent calyx lobes, was clearly shown in 3D models PH223 and PH267. The genuine colors of the mature fruits are also reflected on the 3D models, which are not mentioned in the Flora. On the other hand, the seed descriptions of both species are absent in the Flora. However, through the reconstruction of 3D models PH224 and PH268, their seeds are now visualized which show the differences in size and

shape. This example suggested that 3D models reconstructed by this method not only reflect the morphology of carpological parts as described in the Flora, but also complement the missing information in Flora, which is beneficial to plant authentication and species discovery.

It is noted that a majority of plant species were described after a lag period of 10 to 35 years between specimen collection and publication [42, 43]. The color and the shape of the carpological parts would change and degrade after the lag period, which may affect the judgement of taxonomists. Therefore, documenting the carpological parts as 3D models at their prime stage to preserve the important morphological characters could potentially provide more information for taxonomists at the time of description. This information is extremely crucial when it comes to the description of holotype and the selection of lectotype, hence, improving the accuracy for authentication and species discovery.

In addition, the 3D models reconstructed by this platform could help witness the morphological changes of the carpological structures at different time points, that is especially important in studying fruit maturation. Basically, the structural or caducous changes in various stages of drying fruits, such as detachment of stalk or calyx and even exocarp rupture would be recorded. Moreover, the seed dispersal process could also be documented in the 3D models. For instance, a series of 3D models of the fruit of *Lophostemon confertus* (R. Brown) Peter G. Wilson & J. T. Waterhouse (PH054A, PH054B, PH054C, PH054D) were reconstructed to record its morphological changes on day 1, day 2, day 5 and day 10 respectively (Fig 11). On day 1, the fruit was not yet opened, enclosing all seeds. On day 2, the valves on the top of the fruit opened and started to disperse some seeds. On day 5, majority of the seeds had been released, and the 3-lobed capsule could be easily observed from the 3D model. On day 10, all the seeds had been released, and the wrinkles on the fruit surface become prominent. To conclude, the 3D models could provide significant evidences in continuous assessment of fruit development under specific time points.

Each 3D model beyond PH056 is linked to an authentic specimen deposited in our herbarium, which recorded the carpological part of the specimen in its original state. This provides additional information of the voucher specimen for any possible research study now and in the future. By comparing the 3D models of carpological parts from multiple individuals of the same species with their genetic differences, the results may give better visualization and understanding on gene and phenotypic structure. The 3D models may also be used to visualize the morphological changes of fruits and seeds of species with wide distribution range to study their adaptation to different geographical conditions. The virtual carpological herbarium can further become an online database displaying different types of fruits and seeds, which could be beneficial to studies related to seed dispersal, forest restoration, and animals' diet.

The above examples shown that this method is able to improve plant knowledge by documenting the carpological materials into 3D models. The 3D models recorded the genuine color and the original shape of the carpological materials that are usually lost or changed in traditional specimens due to specimen processing or degradation. This provides supporting information during specimen inspection to reflect the important morphological characteristics in their prime condition for plant authentication and possibly new species discovery. In addition, by reconstructing 3D models of carpological materials at consecutive stages, color changes and structural changes in all dimensions can be recorded, which gives better understandings on fruit and seed development and their dispersal. Since the 3D models are virtual data which can be transferred through internet, together with the user-friendly interface, plant co-workers and outside users can inspect the carpological parts from all angles without physical limitation, which facilitate the exchange of plant knowledge and information around the world.

This research study targets on the documentation of plant carpological parts, but once this technique becomes more advanced in the future, it can be applied on the 3D documentation

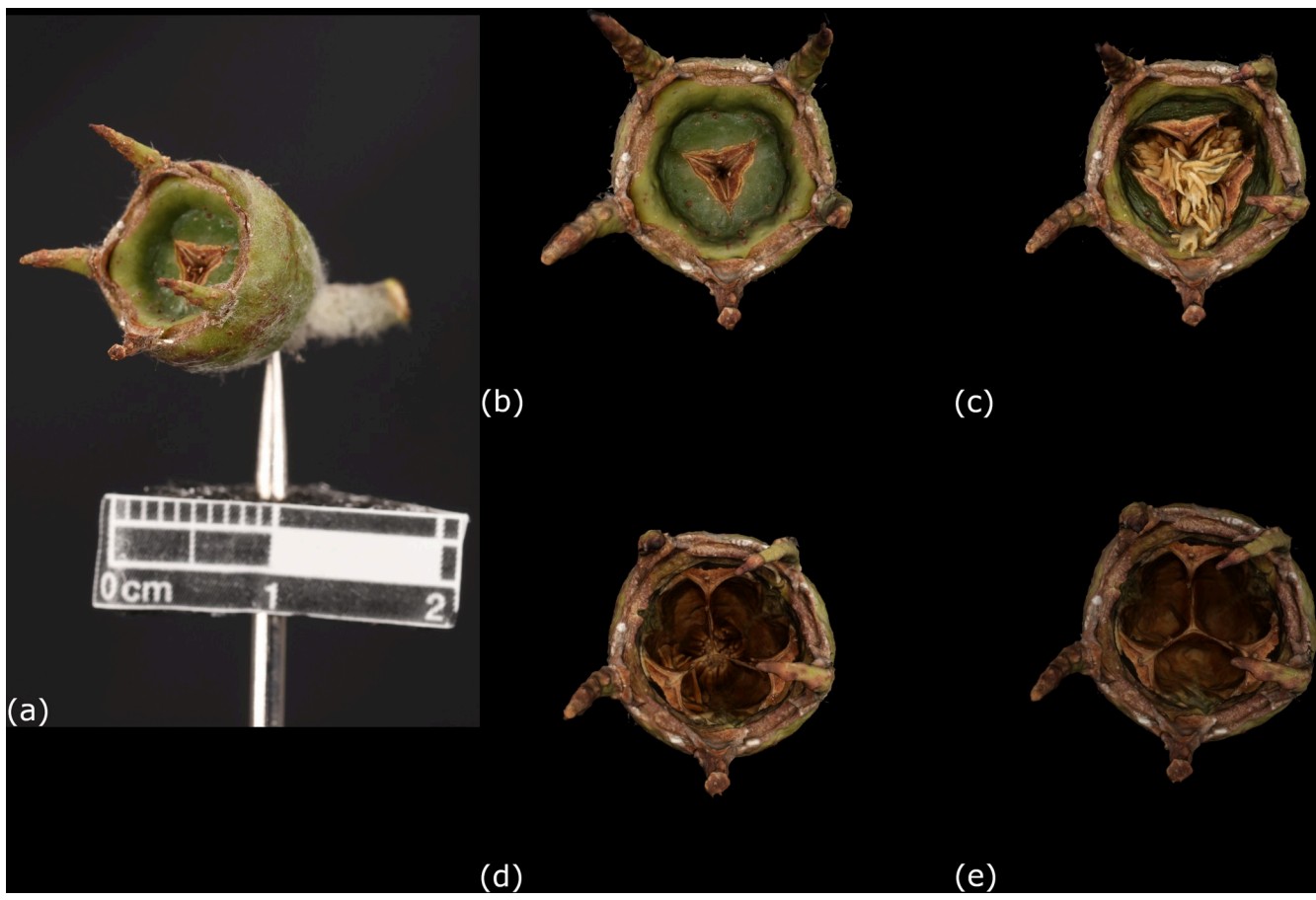

**Fig 11. 3D models of *Lophostemon confertus* (R. Brown) Peter G. Wilson & J. T. Waterhouse at different time points after collection.** (a) A carpological material of the fruit of *L. confertus* was fixed for image capturing. (b) Top view of the 3D model (PH054) reconstructed on day 1. The valves on top of the fruit were closed. (c) Top view of the 3D model (PH054) reconstructed on day 2. The valves of the capsule started to retreat, and the seeds inside were exposed. Two sepals contracted and bended inwards. (d) Top view of the 3D model (PH054) reconstructed on day 5. Most of the seeds had fallen and the internal structure of the capsule was clearly seen. (e) Top view of the 3D model (PH054) reconstructed on day 10. All the seeds were released.

of other plant components such as leaves, flowers or even the whole plants, which is beneficial in the development of a virtual herbarium.

## Future perspective from this research

The photogrammetric platform in this study is beneficial for the development of a virtual herbarium or museum with additional carpological information. A new database of more than 250 3D models of carpological collection was created. This study not only built up a good capacity of documenting a single seed or fruit, but also well recorded some infructescence, e.g. *C. lamontii* (PH045). This 3D model database could be incorporated as important archive of an herbarium. Hence, these plant specimens and carpological materials will no longer be only limited to professionals and researchers, but also be accessible and useful to the general public.

The deliverables from this project had contributed to the establishment of "Virtual Carpological Herbarium of Fruits and Seeds" in the Shiu-Ying Hu Herbarium (https://syhuherbarium. sls.cuhk.edu.hk/collections/3d-specimen/). All information of the database is presented with user-friendly interfaces in all desktop computers and mobile devices. It is expected that the database will facilitate basic science research and its application in plant authentication, species discovery and conservation.

## Supporting information

**S1 Table. Carpological materials adopted in this study for 3D model reconstruction.** The carpological specimens no., species name, carpological parts, size and voucher specimens no. (if any) of the carpological material were documented.
(DOCX)

## Author Contributions

**Conceptualization:** Ho Lam Wang, Tin Hang Wong, Yiu Man Chan, David Tai Wai Lau.

**Data curation:** Ho Lam Wang, Tin Hang Wong, Yat Sum Cheng.

**Formal analysis:** Ho Lam Wang.

**Funding acquisition:** David Tai Wai Lau.

**Investigation:** Ho Lam Wang, Tin Hang Wong, Yiu Man Chan.

**Methodology:** Ho Lam Wang, Tin Hang Wong, Yiu Man Chan, Yat Sum Cheng.

**Project administration:** Tin Hang Wong, David Tai Wai Lau.

**Resources:** Yiu Man Chan.

**Software:** Ho Lam Wang, Tin Hang Wong, Yiu Man Chan, Yat Sum Cheng.

**Supervision:** David Tai Wai Lau.

**Validation:** Ho Lam Wang, Tin Hang Wong, David Tai Wai Lau.

**Writing – original draft:** Ho Lam Wang.

**Writing – review & editing:** Tin Hang Wong, Yiu Man Chan, Yat Sum Cheng, David Tai Wai Lau.

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
