## [Decision Letter · Decision Letter 0]

24 Mar 2022

PONE-D-22-00891Photogrammetric reconstruction of 3D carpological collection in high resolution for plants authentication and species discoveryPLOS ONE

Dear Dr. LAU,

Thank you for submitting your manuscript to PLOS ONE. After careful consideration, we feel that it has merit but does not fully meet PLOS ONE’s publication criteria as it currently stands. Therefore, we invite you to submit a revised version of the manuscript that addresses the points raised during the review process.

See below the comments of your paper and provide a point by point answer.

We look forward to receiving your revised manuscript.

Kind regards,

Antonio Agudo

Academic Editor

PLOS ONE

Journal Requirements:

Additional Editor Comments (if provided):

Dear authors,

Both reviews consider the paper to contain good points for publication. However, both of them still have several concerns to be addressed. Please address the comments and requests point by point. In addition, I think the authors should provide additional comparisons with competing techniques by including some quantitative evaluations, even in simple cases (or synthetic) could be enough. Regarding failure cases, I think they need to be deeply shown and discussed in the paper to help the reader. What about the use of photometric stereo approaches? I believe current methods could generate very accurate estimations in this context. Evaluation of accuracy from a theoretical point of view. Why should this method produce better solutions?

Best

Reviewers' comments:

Reviewer's Responses to Questions

**Comments to the Author**

1. Is the manuscript technically sound, and do the data support the conclusions?

Reviewer #1: Yes

Reviewer #2: Yes

2. Has the statistical analysis been performed appropriately and rigorously? 

Reviewer #1: N/A

Reviewer #2: Yes

3. Have the authors made all data underlying the findings in their manuscript fully available?

Reviewer #1: Yes

Reviewer #2: Yes

4. Is the manuscript presented in an intelligible fashion and written in standard English?

Reviewer #1: No

Reviewer #2: Yes

5. Review Comments to the Author

Reviewer #1: This is a very interesting and potentially very useful contribution for the creation of 3D images of plant fruits and seeds. I believe that those interested in creating a collection of such images would find the information provided here very useful. For those who might be trying to decide whether or not to embark on the creation of such an image library, as part of a virtual herbarium, or some other project, it would be useful to have some additional information -- what is the image file size generated, what is the cost, or amount of time needed to capture a completed image. I think the arguments presented for why one should create such an imaging library could be strengthened -- how are such images used in research today, both within the plant biodiversity community, and by outside users? What are the key characteristics that can be observed through this method that cannot be seen with traditional 2D images? If the creation of a carpological 3D image library should indeed be a key component of a virtual herbarium, as the authors suggest, then there needs to be more of a discussion of how such images improve plant knowledge.

Reviewer #2: This paper provides an accurate and efficient method to reconstruct detailed and high-resolution digital 3D models of carpological materials by photogrammetric method. It gives the method in details on selection of digital camera body, camera lens, adjustment of light source, and software selection. It is very like a guideline of 3D model reconstruction. And the readers can get the related information from other places, such as the guide of the 3D model software. But I think the paper should focus on how to do plants authentication and species discovery using the reconstructed 3D model. Hopefully, the authors can add the above related content.

6. PLOS authors have the option to publish the peer review history of their article (what does this mean?). If published, this will include your full peer review and any attached files.

Reviewer #1: **Yes: **Barbara M Thiers

Reviewer #2: No

---

## [Author Response · Author response to Decision Letter 0]

3 May 2022

Reply to reviewer 1: Thank you very much for your appreciation. We agreed with your suggestions and advice. Amendments has been done in the manuscript. 

Reply to reviewer 2: Thank you very much for your appreciation. We agreed with your suggestions and advice. Further discussion has been added in the manuscript.

---

## [Decision Letter · Decision Letter 1]

7 Jun 2022

Photogrammetric reconstruction of 3D carpological collection in high resolution for plants authentication and species discovery

PONE-D-22-00891R1

Dear Dr. LAU,

We’re pleased to inform you that your manuscript has been judged scientifically suitable for publication and will be formally accepted for publication once it meets all outstanding technical requirements.

Kind regards,

Antonio Agudo

Academic Editor

PLOS ONE

Additional Editor Comments (optional):

Dear authors,

Thank you very much for providing a point-by-point review. Both reviewers considered the paper ready for publication.

Best

Reviewers' comments:

Reviewer's Responses to Questions

**Comments to the Author**

1. If the authors have adequately addressed your comments raised in a previous round of review and you feel that this manuscript is now acceptable for publication, you may indicate that here to bypass the “Comments to the Author” section, enter your conflict of interest statement in the “Confidential to Editor” section, and submit your "Accept" recommendation.

Reviewer #1: All comments have been addressed

Reviewer #2: All comments have been addressed

2. Is the manuscript technically sound, and do the data support the conclusions?

Reviewer #1: Yes

Reviewer #2: Yes

3. Has the statistical analysis been performed appropriately and rigorously? 

Reviewer #1: N/A

Reviewer #2: Yes

4. Have the authors made all data underlying the findings in their manuscript fully available?

Reviewer #1: Yes

Reviewer #2: Yes

5. Is the manuscript presented in an intelligible fashion and written in standard English?

Reviewer #1: Yes

Reviewer #2: Yes

6. Review Comments to the Author

Reviewer #1: (No Response)

Reviewer #2: The paper has been revised according to my suggestions and I think it can be accepted. This paper gives very detailed guide for 3D model reconstruction and comparison with different methods.

7. PLOS authors have the option to publish the peer review history of their article (what does this mean?). If published, this will include your full peer review and any attached files.

Reviewer #1: **Yes: **Barbara Mary Thiers

Reviewer #2: No

---

## [Editor Report · Acceptance letter]

21 Jul 2022

PONE-D-22-00891R1 

Photogrammetric reconstruction of 3D carpological collection in high resolution for plants authentication and species discovery 

Dear Dr. LAU:

I'm pleased to inform you that your manuscript has been deemed suitable for publication in PLOS ONE. Congratulations! Your manuscript is now with our production department. 

Kind regards, 

on behalf of

Dr. Antonio Agudo 

Academic Editor

PLOS ONE